# QUERY-AWARE LEARNABLE GRAPH POOLING TOKENS AS PROMPT FOR LARGE LANGUAGE MODELS

## ABSTRACT

Graph-structured data plays a vital role in numerous domains, such as social networks, citation networks, commonsense reasoning graphs and knowledge graphs. While graph neural networks have been employed for graph processing, recent advancements have explored integrating large language models for graph-based tasks. In this paper, we propose a novel approach named Learnable Graph Pooling Token (LGPT), which addresses the limitations of the scalability issues in node-level projection and information loss in graph-level projection. LGPT enables flexible and efficient graph representation by introducing learnable parameters that act as tokens in large language models, balancing fine-grained and global graph information. Additionally, we investigate an Early Query Fusion technique, which fuses query context before constructing the graph representation, leading to more effective graph embeddings. Our method achieves a 4.13% performance improvement on the GraphQA benchmark without training the large language model, demonstrating significant gains in handling complex textual-attributed graph data.[1]

## 1 INTRODUCTION

A graph is a data structure composed of nodes and edges that represent the relationships between those nodes. Graphs are essential for representing complex relational situations in the real world. For example, *social networks* (Li et al., 2024; Myers et al., 2014) like X (Twitter) and urban networks, *citation networks* (Hu et al., 2020) in academic fields that represent authorship, affiliations, and citations, *protein and molecular graphs* (Cao et al., 2023) for depicting complex molecular interactions, *commonsense reasoning graphs* like ConceptNet (Speer et al., 2017), and *knowledge graphs* such as Wikidata (Vrandečić & Krötzsch, 2014) that store various facts. Traditionally, graph data has been processed using handcrafted feature extraction methods like Katz Index and PageRank (Katz, 1953; Page, 1999). However, with the recent advancements in deep learning, Graph Neural Networks (GNNs) such as GCN, GAT, and Graph Transformer have become widely researched for processing graphs (Kipf & Welling, 2016; Veličković et al., 2017; Shi et al., 2020).

Meanwhile, the field of Natural Language Processing (NLP) has experienced a revolutionary shift with the advent of Large Language Models (LLMs). These models, pre-trained on massive text datasets, possess general problem-solving abilities (Chung et al., 2024). Recently, it has been reported that LLMs can also understand the structural information of graphs and solve graph tasks (Fatemi et al., 2023; Wang et al., 2024b; Chai et al., 2023). The combination of LLMs and graphs is particularly useful in Text-Attributed Graphs (TAGs), where each node and edge contains textual features. One of the simplest approaches is to transform graph information into text and feed it into the LLM through knowledge-augmented prompting, as demonstrated by methods like Baek et al. (2023) and Wu et al. (2023).

However, graphs contain highly complex structural information, making it difficult to convert them into text. Moreover, the performance varies significantly depending on how the graph is textualized, and the optimal text encoding method is still unknown (Fatemi et al., 2023). To overcome these limitations, Perozzi et al. (2024) has achieved significant performance improvements by embedding graph data using GNNs and projecting it into the word embedding space of LLMs through contin-

---

[1]We release our code and materials at `https://github.com/iclr-lgpt/LGPT`

uous prompting. Furthermore, Tian et al. (2024) proposed a technique that distills query-related information during the interaction between graph and query via cross-modality pooling.

This continuous prompting method for graphs can be categorized into node-level projection and graph-level projection (Ren et al., 2024). Node-level projection passes the information of all nodes, obtained through the GNN, to the LLM and is used in tasks such as node classification or link prediction, which require fine-grained structural information. Graph-level projection compresses node representations into a single vector and passes it to the LLM, which is useful in tasks like graph classification that require global graph information.

However, both approaches have limitations. In node-level projection, each node representation is treated as a token by the LLM. Since graphs tend to grow exponentially, this method lacks scalability given the limited prompt length of LLM. Even if a model, like Beltagy et al. (2020), can process extremely long prompts, the computational cost becomes prohibitive. Graph-level projection, where all node information is pooled into a single vector and passed to the LLM, avoids the scalability issue. However, converting a graph with complex context into a single vector results in information loss (Bahdanau, 2014). Given that LLMs must process a graph with vast amounts of information as a single token, this is inevitable.

To address these limitations, we propose a new concept named Learnable Graph Pooling Token (LGPT). This allows graph information to be represented by $n$ learnable parameters, which are passed to the LLM as $n$ tokens. This approach resolves both the computational issue of node-level projection and the information loss problem in graph-level projection. Additionally, we investigate an early query fusion method and deal with the limitations of a late query fusion method. While cross-modality pooling as the late query fusion in Tian et al. (2024) combines query context with graph embeddings, it does so after the graph is encoded. In contrast, we propose an approach that integrates query context before constructing the graph representation, thereby offering a more effective graph embedding method that takes the query context into account.

Our main contribution is as follows:

- We propose a novel concept of Learnable Graph Pooling Token (LGPT), which enables balanced projection between node-level and graph-level projection. As a result, we achieved more than a 4.13% improvement in performance on the GraphQA benchmark dataset without LLM training.

- We explore a method to integrate the early query fusion method during the graph embedding process. Through experiments, we demonstrate that incorporating query context before constructing the node embeddings of the graph leads to greater performance improvements than combining it afterward.

## 2 RELATED WORKS

### 2.1 LLM AS GRAPH ENCODER

Fatemi et al. (2023) and Wang et al. (2024b) demonstrated that encoding graphs into various textual forms allows LLMs to solve graph-centric tasks. Additionally, Wang et al. (2024a) advanced Chain of Thought (CoT) (Wei et al., 2022) into a graph-suitable format by adding the instruction "Let's construct a graph with the nodes and edges first" enabling LLMs to map graph information into conceptual space. However, these approaches all have the limitation of processing graphs at the text level. Since graphs contain complex relational information, converting them into text loses a lot of structural knowledge. To overcome these limitations, Perozzi et al. (2024); Tian et al. (2024) have emerged that embed graphs using GNNs and integrate them with LLMs. Notably, He et al. (2024) achieved significant performance improvements by using both textual graphs and GNN embeddings.

### 2.2 LEARNABLE POOLING METHOD

Sum and Mean Pooling have traditionally been used as readout functions to create graph embeddings from node embeddings in GNNs. However, they suffer from scalability issues that arise when dealing with graphs that have a varying number of nodes, an inability to emphasize important nodes

and information loss in compressing node information into a single vector. To address these limitations, learnable pooling methods that incorporate learnable parameters have been explored. Ying et al. (2018) introduced hierarchical pooling by applying soft clustering to reflect the hierarchical structure of graphs. Also, Lee et al. (2019) proposed a learnable pooling method by incorporating an attention mechanism to capture more information from important nodes. Nevertheless, these approaches still face the risk of information loss as they condense numerous node embeddings into a single graph embedding vector (Bahdanau, 2014).

## 2.3 QUERY AWARE GRAPH REPRESENTATION

In Tian et al. (2024), a method was introduced to combine graph and query representations as cross modality pooling using a cross-attention mechanism. While this merges the information from the graph and the query, it has the limitation of being a *Late Fusion* approach, as the graph and query information are encoded independently before being combined. In Yasunaga et al. (2021), a virtual query node is created within the graph to perform graph encoding that is dependent on the meaning of the query. This approach is effectively used in *Early Fusion* for query-aware graph representation, as seen in its connection to instruction nodes in models like Zhang et al. (2022) and Yasunaga et al. (2022).

# 3 METHODOLOGY

## 3.1 PROBLEM STATEMENT

We address the problem of Textual Attributed Graph Question Answering (Graph QA) by combining a graph encoder with a large language model. In Graph QA tasks, a query $q$ and a graph $\mathcal{G}$ which is provided as external knowledge related to the query are given. Our goal is to generate the optimal answer $a^*$ for $q$ by utilizing the information contained in $\mathcal{G}$.

$$a^* = \arg\max_a p(a|q, \mathcal{G}) \tag{1}$$

In this context, $\mathcal{G}$ is a text-attributed graph, where both nodes and the edges are associated with textual attributes. Formally, $\mathcal{G}$ is defined as $\mathcal{G} = \{\mathbb{V}, \mathbb{L}, \{\boldsymbol{x}_n\}_{n\in\mathbb{V}}, \{\boldsymbol{x}_l\}_{l\in\mathbb{L}}\}$, where $\mathbb{V}$ and $\mathbb{L}$ represent the sets of nodes (vertices) and edges (links). $\boldsymbol{x}_n$ and $\boldsymbol{x}_l$ denote the textual attributes of the nodes and edges.

The $p(a|q, \mathcal{G})$ is composed of a Graph Retriever $p_\theta(\mathcal{S}|q, \mathcal{G})$ and an Answer Generator $p_\phi(a|q, \mathcal{S})$ where $\mathcal{S}$ is the sub-graph which related with $q$ (Peng et al., 2024). In this paper, we borrow He et al. (2024) results for the graph retriever and focus on optimizing $p_\phi(a|q, \mathcal{S})$ by redefining the problem.

$$p(a|q, \mathcal{G}) = p_\theta(\mathcal{S}|q, \mathcal{G})p_\phi(a|q, \mathcal{S}) \tag{2}$$
$$\approx p_\phi(a|q, \mathcal{S}) \tag{3}$$

## 3.2 QUERY AWARE LEARNABLE GRAPH POOLING TOKENS

### 3.2.1 OVERVIEW

The process of $p_\phi(a|q, \mathcal{S})$ is divided into three main components as shown in Figure 1. First, the given sub-graph $\mathcal{S}$ is transformed into a textual graph via a discrete prompt template $T$. Then it is processed by word embedding layer WE of the LLM. Second, the graph $\mathcal{S}$ is converted into graph embeddings through a graph encoder $\text{GE}_\psi$ which has learnable parameters $\psi$.

$$\boldsymbol{E}_{text} = \text{WE}(T(\mathcal{S})) \quad where, \boldsymbol{E}_{text} \in \mathbb{R}^{|text| \times d} \tag{4}$$
$$\boldsymbol{E}_{\mathcal{S}} = \text{GE}_\psi(\mathcal{S}) \quad where, \boldsymbol{E}_{\mathcal{S}} \in \mathbb{R}^{n \times d} \tag{5}$$

The discrete prompt embedding $\boldsymbol{E}_{text}$ and the continuous prompt embedding $\boldsymbol{E}_{\mathcal{S}}$ are concatenated and input into LLM along with the query $\text{WE}(q)$ which is processed by WE. Our objective is to optimize the word distribution of the predicted answer $a$, aligning it with the word distribution of the optimal answer $a^*$. To this end, both the LLM and WE are frozen in their pre-trained states,

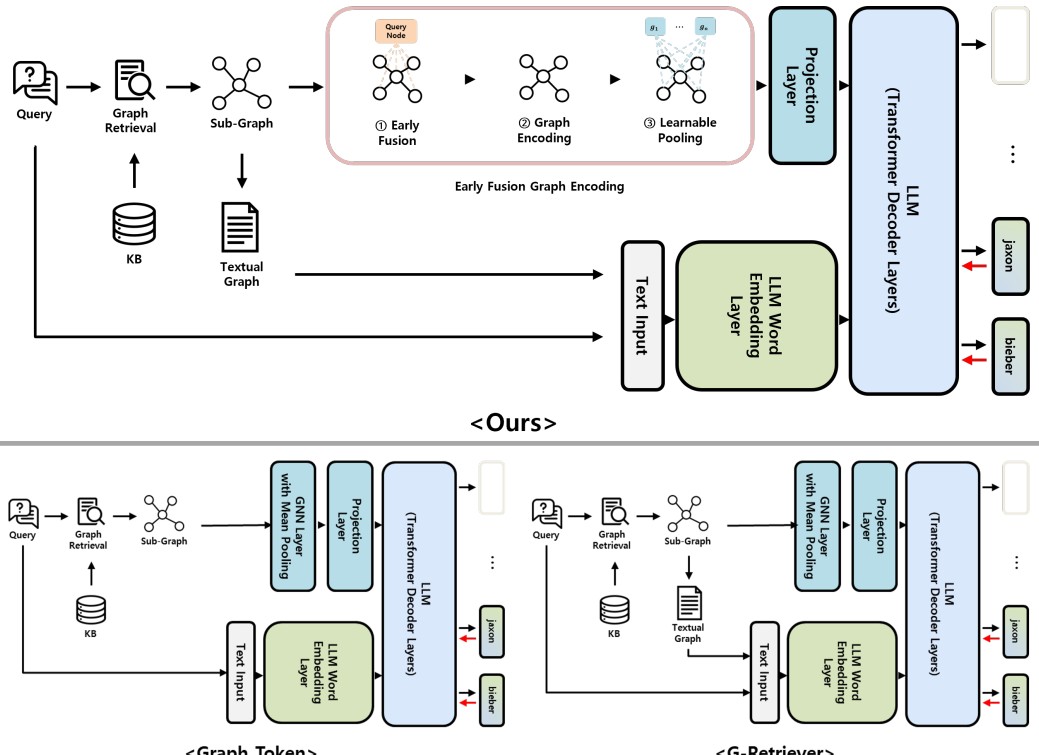

Figure 1: **Overview of Proposed Method.** Our approach is similar to Perozzi et al. (2024); He et al. (2024). Graph Token (Perozzi et al., 2024) generates node embeddings from the given graph $\mathcal{S}$ using a GNN encoder and applies mean pooling to deliver the graph information to the LLM. G-Retriever (He et al., 2024) follows the same process but differs in that it transforms the given graph $\mathcal{S}$ into a textual graph and feeds it into the LLM along with the additional information. Our approach builds on G-Retriever by incorporating LGPT and an Early Query Fusion Module (Red Box).

while we focus on optimizing the $GE_\psi$. In equations (4), (5) and (6), $d$ represents the dimension of the embedding vector, $|text|$ and $|q|$ refer to the number of tokens in the tokenized text and query. $n$ refers to the number of learnable graph pooling tokens, which is described in section 3.2.3.

$$p_\phi(a|q, \mathcal{S}) = \text{LLM}([\boldsymbol{E}_{\mathcal{S}}; \boldsymbol{E}_{text}; \text{WE}(q)])$$
$$where, [\boldsymbol{E}_{\mathcal{S}}; \boldsymbol{E}_{text}; \text{WE}(q)] \in \mathbb{R}^{(n+|text|+|q|) \times d} \quad (6)$$

### 3.2.2 EARLY QUERY FUSION

The total amount of information $I_{total}$ contained in the given graph $\mathcal{S}$, has exponential complexity because a graph represents relationships between nodes. However, for our goal, it is not necessary to utilize all of the whole information. Instead, only the information relevant to the query $I_{query}$ which is much smaller than $I_{total}$, is sufficient. Note that, $I_{total} \gg I_{query}$.

In Tian et al. (2024), the approach first embeds the graph with exponential complexity and then filters $I_{query}$ through cross-modality pooling. This leads to ineffective, as the entire graph with exponential complexity is encoded first. In contrast, we enhance the effectivity of information representation by adopting an early fusion method, where query information is fused before the graph embedding is generated. To achieve this, the query is embedded in the graph embedding space as a virtual query node $\boldsymbol{n}_q$ using text encoder TextEnc (Yasunaga et al., 2021).

$$\boldsymbol{n}_q = \text{TextEnc}(q) \quad (7)$$

The query node $\boldsymbol{n}_q$ connects all nodes in the graph $\mathcal{S}$. It performs message passing using $\text{GNN}_{query}$, resulting in the graph $\mathcal{S}_q$ that incorporates the query node embedding $\boldsymbol{n}'_q$ and the original node embeddings. Subsequently, $\text{GNN}_{graph}$ is employed to encode the original relational information of the graph $\mathcal{S}_q$.

$$\{\boldsymbol{n}'_q\} \cup \{x_n\}_{n \in \text{nodes of } \mathcal{S}_q} = \text{GNN}_{query}(\mathcal{S}, \boldsymbol{n}_q) \tag{8}$$

$$\{x_n\}_{n \in \text{nodes of } \mathcal{S}_g} = \text{GNN}_{graph}(\mathcal{S}_q) \tag{9}$$

### 3.2.3 LEARNABLE GRAPH POOLING TOKENS (LGPT)

There are two main approaches to prompting with a graph encoder. The first approach involves passing all node embeddings to the LLM, while the second approach uses a readout function to transform node embeddings into single graph embedding, which is then passed to the LLM. In node-level prompting, each node is treated as a token by the LLM, but as the number of nodes increases, this method becomes impractical due to scalability issues. On the other hand, in graph-level prompting, methods such as mean pooling, DiffPool(Ying et al., 2018) and SAGPool (Lee et al., 2019) are used to compress node embeddings into a single vector, which is then provided to the LLM. However, this requires encoding all the graph information into a single vector, increasing the risk of information loss (Bahdanau, 2014).

To address this issue, we propose a novel pooling method named Learnable Graph Pooling Tokens (LGPT). LGPT employees $n$ learnable parameters with the same dimension as the node embeddings, which are fully connected to all the nodes in the given graph $\mathcal{S}_g$. After that, message passing is performed through a $\text{GNN}_{pool}$ process, resulting in $\mathcal{S}_p$ and graph representation $n$ tokens $\{\boldsymbol{g}'_1 \cdots \boldsymbol{g}'_n\}$. This method reduces the risk of information loss compared to previous approaches that represented the graph using only a single vector. Finally, processed LGPT $\{\boldsymbol{g}'_1 \cdots \boldsymbol{g}'_n\}$ is transformed into $\boldsymbol{E}_{\mathcal{S}}$, the input format for the LLM, through a projection layer Proj composed of Multi-Layer Perceptron (MLP).

$$\text{LGPT} : \{\boldsymbol{g}_1, \cdots, \boldsymbol{g}_n\} \tag{10}$$

$$\{\boldsymbol{g}'_1 \cdots \boldsymbol{g}'_n\} \cup \{x_n\}_{n \in \text{nodes of } \mathcal{S}_p} = \text{GNN}_{pool}(\mathcal{S}_g, \{\boldsymbol{g}_1, \cdots, \boldsymbol{g}_n\}) \tag{11}$$

$$\boldsymbol{E}_{\mathcal{S}} = \text{Proj}(\{\boldsymbol{g}'_1 \cdots \boldsymbol{g}'_n\}) \tag{12}$$

The reason LGPT works effectively is that it conceptually combines two learnable pooling methods. Ying et al. (2018) performs pooling hierarchically through soft clustering, where a node can be assigned to multiple clusters. In our method, since all nodes are connected to learnable tokens and perform message passing for pooling, each LGPT token can be seen as a soft cluster, making our approach conceptually aligned with soft clustering in Ying et al. (2018). Additionally, by using Graph Transformer (Shi et al., 2020) as the GNN architecture, our method operates similarly to Lee et al. (2019), which employs the self-attention mechanism. In essence, our method works well because it conceptually borrows from both Ying et al. (2018) and Lee et al. (2019). However, the key difference from these methods is that, instead of pooling into a single graph embedding, our approach uses multiple learnable tokens for pooling, thereby reducing information loss.

In summary, the proposed method consists of two main components: Early Query Fusion and the Learnable Graph Pooling Tokens. Learnable parameters of $\text{GNN}_{query}$ and $\text{GNN}_{graph}$ are required for Early Query Fusion while learnable parameters of $\text{GNN}_{pool}$ and LGPT are necessary for the graph pooling.

$$\psi = \text{parameters of } \{\text{GNN}_{query}, \text{GNN}_{graph}, \text{GNN}_{pool}, \text{LGPT}, \text{Proj}\} \tag{13}$$

### 3.3 ANALYSIS OF TIME COMPLEXITY

Let $k$ denote the number of nodes, $t$ the number of prompt text tokens, $g$ the number of GNN layers, and $n$ the number of LGPTs. The time complexity of the proposed Graph Encoder, which employs three GNNs, is thus $O(3g(n + k))$. In comparison, the time complexity of both G-Retriever and GraphToken is $O(gk)$. Given that $n \ll k$, the time complexity of our method aligns with that of other baseline graph encoders, remaining at $O(gk)$.

The computational time complexity for the LLM component is determined by the self-attention mechanism and is proportional to the square of the prompt length. In our model, the prompt consists of $t + n$ tokens, whereas GraphToken and G-Retriever process $t + 1$ tokens. We set $k = 8$ to ensure $n \ll t$, thereby maintaining the time complexity of LLM computation in our model at $O(t^2)$, consistent with that of other baseline models.

## 4 EXPERIMENTS

### 4.1 EXPERIMENT SETUP

#### 4.1.1 DATASETS

For our experiments, we used the GraphQA benchmark dataset (Table 1) released by He et al. (2024). The experiments are conducted using the QA data and the corresponding graphs provided. This dataset consists of three sub-QA tasks as follows. The datasets are divided into train, validation and test subsets using a 6:2:2 ratio.

- **ExplaGraphs (Saha et al., 2021)**: The dataset is a commonsense reasoning dataset composed of 2,766 graphs for stance prediction in debates. The task is evaluating whether two arguments support or not to use the information from the given graph. The evaluation metric used is accuracy.

- **SceneGraphs**: The SceneGraphs dataset, derived from GQA (Hudson & Manning, 2019), contains 100,000 scene graphs detailing objects, attributes and relationships within images. It challenges users with tasks requiring spatial understanding and multi-step inference to answer open-ended questions based on scene graph descriptions, evaluated on accuracy.

- **WebQSP (Yih et al., 2016; Luo et al., 2023)**: WebQSP is a large-scale knowledge Graph QA dataset with 4,737 questions, requiring multi-hop reasoning to answer. It uses a subset of Freebase, containing facts within 2-hops of the entities in the questions. The task is evaluated using the Hits@1 metric to measure the precision of the top answer.

| Dataset | ExplaGraphs | SceneGraphs | WebQSP |
|---|---|---|---|
| # Graph | 2766 | 100000 | 4737 |
| Avg. # Nodes | 5.17 | 19.13 | 1370.89 |
| Avg. # Edges | 4.25 | 68.44 | 4252.37 |
| Node Attribute | Commonsense Concepts | Object Attributes (e.g. color, shape) | Entities in KG (Freebase) |
| Edge Attribute | Commonsense Relations | Relations (e.g. actions, spatial relations) | Relations in KG (Freebase) |
| Task | Commonsense reasoning | Scene graph QA | Knowledge Graph QA |

Table 1: Summary of GraphQA benchmark Dataset He et al. (2024).

#### 4.1.2 IMPLEMENTATION DETAILS

We set our implementation details to be consistent with He et al. (2024) such as discrete prompt template $T$ to textualize the given graph and how to retrieve graph $S$ from $G$. We used LLaMa2-7b[2] (Touvron et al., 2023) with 4-bit NormalFloat quantization (Dettmers et al., 2021; 2024) as LLM and Setence Transformer[3] (Reimers, 2019) as TextEnc. For GNN architecture we employed a Graph Transformer (Shi et al., 2020) with four layers. The model was trained to minimize Cross Entropy Loss for each label's token using the AdamW (Loshchilov, 2017) optimizer with learning late of 1e-4. The number of LGPT was set to 8 and the performance comparison based on the number of LGPT is discussed in the following section 4.4. All experiments were conducted on a single Nvidia A6000 48GB GPU.

### 4.2 MAIN RESULTS

We compared the results of our approach with various methods that rely on prompting without training LLM. First, as discrete prompt baselines without prompt module training (Inference Only),

---

[2]https://huggingface.co/meta-llama/Llama-2-7b
[3]https://huggingface.co/sentence-transformers/all-roberta-large-v11

**<Zero-CoT>**  **<CoT-BAG>**  **<KAPING>**

Figure 2: **Inference Only Method Details.** Zero-CoT (Kojima et al., 2022) adds the prompt "Let's think step by step" utilizing the core concept of Chain of Thought (Wei et al., 2022), to enable LLMs to generate reasoning processes automatically. CoT-BAG (Wang et al., 2024a) adapts this for graph tasks by modifying the prompt to "Let's construct a graph with the nodes and edges first". On the other hand, KAPING (Baek et al., 2023) prompted the information of the given graph as linearized triples.

Table 2: **Main Results.** The table compares the experimental results of various methods, including Inference Only and Frozen LLM w/ PT. Our model (Ours) demonstrated the highest performance, surpassing both Inference Only and Frozen LLM w/ PT approaches. Compared to G-Retriever, our model shows performance improvements ranging from 2.38% to 5.77%, with an average improvement of 4.13%.

|  |  | # of Prompt Tokens | Expla Graphs | SceneGraphs | WebQSP | Average |
|---|---|---|---|---|---|---|
| Inference Only | Zero-Shot | - | 56.50 | 39.74 | 41.06 | 45.77 |
|  | Zero-CoT | - | 57.04 | 52.60 | 51.30 | 53.65 |
|  | CoT-BAG | - | 57.94 | 56.80 | 39.60 | 51.45 |
|  | KAPING | - | 62.27 | 43.75 | 52.64 | 52.89 |
| Frozen LLM w/ PT | Prompt Tuning | 10 | 57.63 | 63.41 | 48.34 | 56.46 |
|  | Graph Token | 1 | 85.08 | 49.03 | 57.05 | 63.72 |
|  | G-Retriever | 1 | 85.16 | 81.31 | 70.49 | 78.99 |
|  | Ours | 8 | **90.07** | **84.50** | **72.17** | **82.25** |
|  | Δ G-Retriever |  | +5.77% | +3.92% | +2.38% | +4.13% |

we used zero-shot, Zero-CoT (Kojima et al., 2022), CoT-BAG (Wang et al., 2024a), KAPING (Baek et al., 2023). Also, for a fair comparison with our method, we included methods with trained prompt module training (Frozen LLM w/ PT), such as Prompt Tuning (Lester et al., 2021), Graph Token (Perozzi et al., 2024) and G-Retriever (He et al., 2024). Details of each method are described in Figure 1 and 2.

Table 2 shows the main results. The prompt module with GNN consistently shows better performance than the only inference setting. This suggests that the structural representation of the graph is more appropriately encoded through GNN embeddings. The lower performance of Prompt Tuning, which only trains learnable parameters without GNNs, compared to Graph Token and G-Retriever, further supports it. When comparing Ours to G-Retriever with all settings identical except for Early Query Fusion and LGPT, our approach achieves performance improvements ranging from 2.38% to 5.77%, with an average improvement of 4.13% across the three datasets. The improvement indicates that Early Query Fusion and LGPT further enhance performance by ensuring that query-specific information is integrated early, reducing the risk of information loss.

We conducted ablation studies to analyze the individual effects and interaction of Early Query Fusion and LGPT. Table 3 shows the results. When applying Early Fusion, we observed an average performance improvement of 2.88%. Although there is a 0.22 performance drop on the WebQSP dataset, the reported standard deviation of G-Retriever's performance due to random seed variation is 1.21, suggesting that the performance drop is not statistically significant (He et al., 2024). Additionally, applying LGPT results in an average performance improvement of 3.87%, indicating that

Table 3: **Result of Ablation Studies.** The table shows the results of analyzing the individual effects of Early Query Fusion and Learnable Graph Pooling Tokens (LGPT). Applying Early Query Fusion resulted in an average performance improvement of 2.88%, while LGPT contributed an average improvement of 3.87%. When both methods are applied together, additional performance gains are observed, leading to a total improvement of 4.13% compared to G-Retriever.

| | # of Prompt Tokens | Expla Graphs | SceneGraphs | WebQSP | Average |
|---|---|---|---|---|---|
| G-Retriever | 1 | 85.16 | 81.31 | 70.49 | 78.99 |
| with Early Query Fusion | 1 | 89.53 | 83.98 | 70.27 | 81.26 |
| ∆G-Retriever | | +5.13% | +3.28% | -0.31% | +2.88% |
| with LGPT | 8 | 88.98 | 85.05 | 72.11 | 82.05 |
| ∆G-Retriever | | +4.49% | +4.60% | +2.30% | +3.87% |
| with LGPT and Early Query Fusion | 8 | 90.07 | 84.5 | 72.17 | 82.25 |
| ∆G-Retriever | | +5.77% | +3.92% | +2.38% | +4.13% |

the traditional pooling method using a single vector incurs information loss and our method offers an effective alternative. Finally, when both methods are applied together, they exhibit a positive interaction, achieving an average performance improvement of 4.13%.

In the previous experiments, we reported results by training only the prompt module, without training the LLM, to independently analyze the effects of the proposed method. Additionally, we conduct experiments where both the LLM and the prompt module are trained together. To efficiently train the LLM, we employ a Low-Rank Adaption (LoRA) (Hu et al., 2021). The trainable parameters, including the prompt module, accounted for only 1.82% of the total parameters.

The experimental results are shown in Figure 3. Our approach shows even greater effectiveness when training the LLM with LoRA. Compared to all baselines, our method, which used LoRA for training the LLM, achieved the highest performance improvements. On average, it demonstrated an 86.78% performance improvement over the non-trained LLM and an 11.48% improvement over the LLM trained with LoRA. Additionally, it outperforms G-Retriever with LoRA, which trained both the LLM and the prompt module, by 3.54%. This indicates that our prompting method with training LLM by using LoRA is highly effective in conveying graph information to the LLM and proves to be superior to other prompting methods.

### 4.3 EFFECT OF EARLY QUERY FUSION

Sun et al. (2018) reported that in the process of knowledge enhancement, *Early Fusion*, where information is combined during the representation creation phase, results in greater performance improvements compared to *Late Fusion*, where embeddings are combined after they have been independently generated. On the other hand, Tian et al. (2024) employed late fusion when combining query and graph information, in which the fusion process only occurs after the graph information has been fully encoded.

In this paper, we adopted Early Query Fusion, where query information is integrated before the graph embeddings are generated. To validate the effectiveness of this approach, we conducted experiments comparing early fusion and late fusion methods. For late fusion, we used the cross modality pooling technique from Tian et al. (2024), while for Early Fusion, we applied the Early Query Fusion strategy proposed in this work.

The results present in Table 4 show the differences between the two methods. In the case that is applied mean pooling as the readout function, late fusion performance decreases compared to when query fusion is not used. This suggests that combining fully processed embeddings from different modalities may work as noise or lead to inefficient information integration. On the other hand, early query fusion shows slight performance improvements, indicating that integrating query information earlier in the process allows for better representation and information fusion within the graph structure. Moreover, when both Early Fusion and Late Fusion are applied together, a greater average performance improvement is observed.

Even when applying LGPT in the readout function, early fusion results in greater performance improvements compared to late fusion. Similar to the case with mean pooling, applying late fusion leads to a slight performance decrease. Moreover, combining both methods also results in a per-

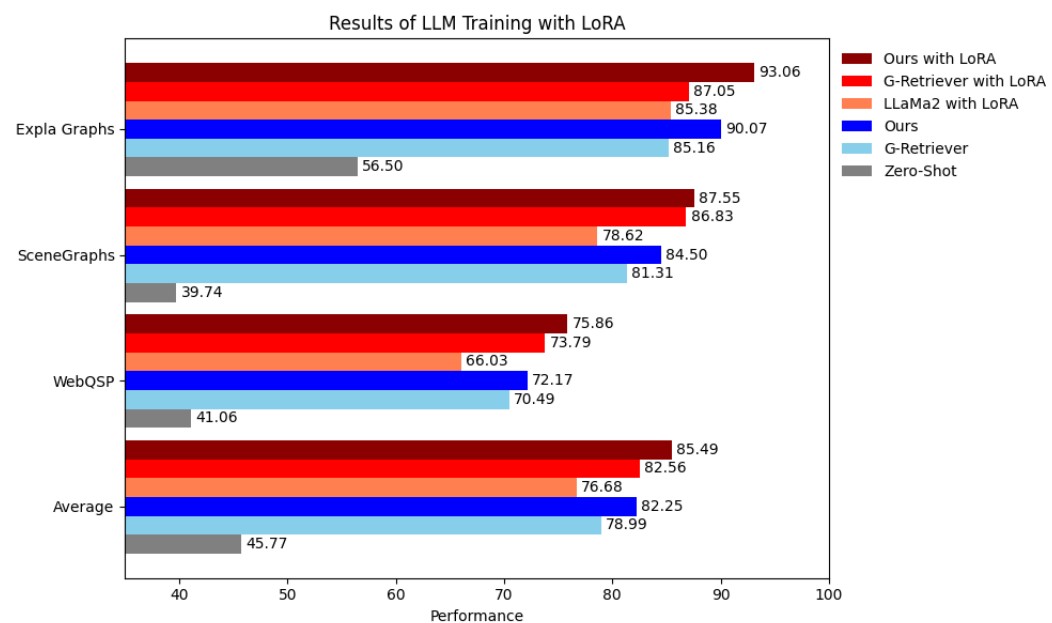

Figure 3: The red bars represent the case where both the LLM and the prompt module were trained using LoRA, while the blue bars represent the case where only the prompt module was trained, and the gray bars represent inference only. Training the LLM using LoRA alongside the prompt module resulted in a significant performance improvement. Additionally, even when training the LLM, our approach, which combines LGPT and the Early Query Fusion Module, demonstrated superior QA performance compared to G-Retriever.

Table 4: **Effect of Query Fusion Method.** The table compares the results of Early Fusion and Late Fusion methods. Applying Early Fusion leads to performance improvements, while Late Fusion results in a decrease in performance. When both methods are applied together, further performance gains are observed than when late fusion is only applied, indicating that Early Fusion is a more effective approach than Late Fusion.

| Readout | # of Tokens | Early Fusion | Late Fusion | Expla Graphs | SceneGraphs | WebQSP | Average |
|---|---|---|---|---|---|---|---|
| Mean Pooling | 1 | X | X | 89.71 | 82.99 | 69.47 | 80.72 |
| | | X | O | 83.75 | 81.20 | 70.51 | 78.49 |
| | | O | X | 89.53 | 83.98 | 70.27 | 81.26 |
| | | O | O | 90.97 | 84.77 | 69.90 | 81.88 |
| LGPT | 8 | X | X | 88.98 | 85.05 | 72.11 | 82.05 |
| | | X | O | 87.36 | 84.35 | 71.56 | 81.09 |
| | | O | X | 90.07 | 84.50 | 72.17 | 82.25 |
| | | O | O | 88.62 | 85.19 | 70.70 | 81.50 |

formance drop. The key takeaway from these experiments is that Early Query Fusion is a more suitable and effective approach for integrating query information with graph structures compared to the traditional Late Fusion method.

## 4.4 PERFORMANCE COMPARISON OF THE NUMBER OF LGPT

In this section, we compared the model's performance on the SceneGraph dataset by varying the number of LGPT from 1, 8, to 32. The results are shown in Figure 4 and regardless of the Query Fusion method, using 8 LGPTs consistently outperforms using just 1 LGPT. This suggests, as mentioned earlier, that encoding the complex information of a graph into a single vector leads to information loss. Specifically, when compressing all the graph information into a single vector, important relationships and characteristics may not be fully captured, resulting in degraded performance.

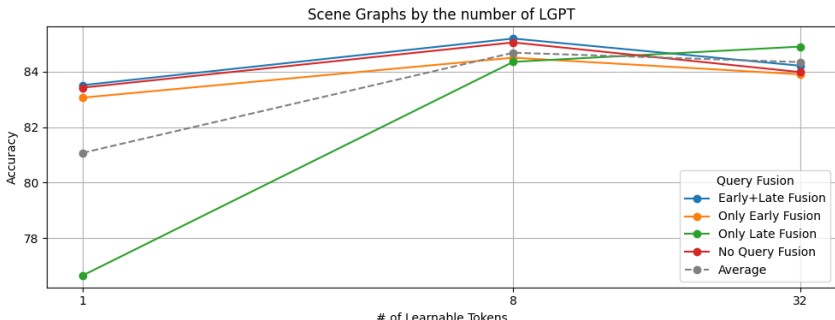

Figure 4: **Performance Comparison of the number of LGPT** The figure presents the performance comparison between Early Fusion and Late Fusion approaches, with varying numbers of Learnable Graph Pooling Tokens (LGPT). The experimental results indicate that using 8 LGPTs yielded the highest performance in both methods, reaching the maximum score for Early Fusion and Late Fusion. However, performance did not improve further when increasing the number of LGPTs to 32, suggesting that beyond a certain point, additional LGPTs do not contribute to further performance gains.

Notably, except for the Late Fusion method, using 8 LGPTs outperform using 32 LGPTs. As the number of learnable parameters increases, the search space during training also expands and having more parameters does not necessarily lead to better performance. Our experimental results support this observation, showing that too many learnable parameters can result in overfitting or information redundancy, which ultimately hinders performance.

However, in the case of the Late Fusion method, performance improves as the number of LGPT increases. This can be attributed to the fact that, in Late Fusion, LGPT is directly involved in the cross-attention operations between the graph and query information. In this context, a greater number of LGPTs allows for richer information exchange, leading to performance gains.

Overall, this experiment highlights the importance of carefully selecting the number of LGPTs. Increasing the number of parameters beyond a certain threshold does not always guarantee performance improvements. In particular, using 8 LGPTs consistently achieves the best performance across different Fusion methods, suggesting that this number strikes a good balance between performance and efficiency.

## 5 CONCLUSION

In this work, we introduced a novel approach, the Learnable Graph Pooling Token (LGPT), which addresses the challenges of graph representation for text-attributed graph question answering tasks. Our method bridges the gap between node-level and graph-level projections by representing graph information with learnable parameters passed as tokens to large language models. This approach mitigates both the scalability issues inherent in node-level projections and the information loss in graph-level projections. Additionally, we proposed an Early Query Fusion technique, which incorporates query information during the graph embedding process, ensuring that query-specific details are embedded into the graph representation before it is constructed. This method demonstrates significant performance improvements over previous approaches using late query fusion.

Through extensive experimentation on the GraphQA benchmark, our approach consistently outperformed existing methods, achieving an average improvement of 4.13% over the baseline model without training LLM. The combination of LGPT and Early Query Fusion proved to be highly effective in addressing the complexities of textual-attributed graphs while ensuring scalable and efficient graph representation without training large language models.

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
