# OpenReview forum: "Query-Aware Learnable Graph Pooling Tokens as Prompt for Large Language Models"
_ICLR.cc/2025/Conference — Submitted to ICLR 2025_

### Official Review · Reviewer_M81S · 2024-10-29

**Soundness:** 2
**Presentation:** 3
**Contribution:** 2
**Rating:** 6
**Confidence:** 3

**Summary:**

- This paper proposes a learnable graph pooling module to enhance LLM-based GraphQA.

**Strengths:**

- The combination of LLM and GNN is an important research topic.
- The design of this paper is reasonable.

**Weaknesses:**

- The novelty seems to be limited in this paper because authors only made a new incremental design in the graph encoder. The core paradigm of graph QA is preserved compared with other baselines.
- Some important GNN+LLM baselines are missing in the experiments. For example, GNP [1].
- The training/inference efficiency of the method should be compared with other baselines.
- The detailed information about the graphs in each dataset is not reported.
- The original dataset and README instructions are not provided in the code, making it difficult to reproduce the performance.




[1] Graph neural prompting with large language models

**Questions:**

- What is the meaning of Sf in the author keywords?
- See weaknesses and make some revisions to the paper.

---

> ### Author Response · Authors · 2024-11-13
>
> Regarding Weakness 1: GNP is a model designed for addressing Multi-Choice QA tasks. Since our task is not a Multi-Choice QA problem, direct comparison with GNP is not feasible. However, we incorporated the core module of GNP, the Cross-Modality Pooling Module, to perform comparative experiments.
>
> ----------
>
> Regarding Weakness 2: Let n denote the number of nodes, t the number of prompt text tokens, g the number of GNN layers, and k the number of LGPTs. The time complexity of our Graph Encoder, which utilizes 3 GNNs, is therefore O(3g(n+k)). Meanwhile, the time complexity of G-Retriever and GraphToken is O(gn). Considering that k << n, the time complexity of our method and that of other baseline graph encoders remain the same at O(gn).
>
> The time complexity required for LLM computation is proportional to the square of the prompt length due to the self-attention mechanism. In our model, t+k tokens are passed in the prompt, while t+1 tokens are passed in GraphToken and G-Retriever. We set k=8 so that k << t, ensuring that the time complexity of LLM computation in our model is identical to that in other baseline models at O(t^2).
>
> ----------
> Regarding Weakness 3: Due to page limitations, we specified the source of our dataset (https://arxiv.org/pdf/2402.07630) as a substitute. However, this is a very relevant point, and we will address this in the Appendix.
>
> ----------
>
> Regarding Weakness 4: We plan to organize the code and make it available on GitHub once the anonymous review period concludes. We apologize for the lack of a comprehensive README file due to concerns about premature exposure of our results. Thank you for your patience.
>
> ----------
>
> Regarding the Question: It appears a typo occurred before submission. We apologize for any confusion this may have caused.
>
> ----------
>
> Your insightful comments have significantly contributed to the improvement of our research. We appreciate the time you dedicated to reviewing our work, and we hope this paper will be accepted to enable further discussions. We kindly ask you to reconsider your evaluation score.

---

> > ### Comment · Reviewer_M81S · 2024-11-20
> > **Thank you for your rebuttal**
> >
> > Dear Authors:
> >
> > Thank you for your response. It address my concerns. I'd like to keep my score.

---

> > > ### Author Response · Authors · 2024-11-26
> > >
> > > We are glad to hear that your concerns have been addressed. Feel free to let us know if you have any further questions; we will happily answer them

---

### Official Review · Reviewer_ENaE · 2024-11-03

**Soundness:** 2
**Presentation:** 3
**Contribution:** 2
**Rating:** 5
**Confidence:** 4

**Summary:**

The paper addresses the problem of Textual-Attributed Graph QA, divided into two main steps: sub-graph retrieval and answer generation. For answer generation, their approach transforms the sub-graph into textual embeddings through a prompt, generates embeddings, and then uses a graph encoder with learnable parameters to process them. The paper highlights scalability issues in node-level prompting (where each node is treated as a separate token in the language model) and information loss in graph-level projection (where the entire graph is compressed into a single vector). To address this, the authors propose Learnable Graph Pooling Tokens (LGPT), a pooling method that introduces learnable parameters (tokens) that connect to all nodes and perform message passing. This method allows for flexible, efficient graph representation that balances fine-grained and global information, achieving improved performance on Graph QA tasks.

**Strengths:**

The paper is easy to read and understand. Extensive experiments and analysis have been shown to prove the proposed method.

**Weaknesses:**

The idea of “early fusion” by forming an external node and fully connecting to other nodes in the graph is not novel to the field. The LGPT idea seems intuitive that increasing the number would increase the performance but would like to see more analysis here.

**Questions:**

1. “ However, the key difference from these methods is that, instead of pooling into a single graph embedding, our approach uses multiple learnable tokens for pooling, thereby reducing information loss” - Is there a pattern in the information loss. Is there a way to quantify this loss other than looking at the accuracy? What kind of data samples perform better when we increase the number of LGPT?

2. How does the number of LGPT performance vary with the different datasets?

---

> ### Author Response · Authors · 2024-11-13
>
> Regarding Weaknesses: Our Early-Fusion approach adopts the methodology of QA-GNN (https://arxiv.org/pdf/2104.06378), as described in the main text. Although it is not a novel concept, we believe our application of this method in conjunction with LLMs represents a significant contribution as the first instance of its kind.
>
> The concern of potential information loss when conveying information through a single parameter has been previously discussed in studies that introduce attention mechanisms to seq2seq models (https://arxiv.org/abs/1409.0473). In our experiments, we quantitatively verified this by comparing the results with LGPT configured with 1, 8, and 32 instances. However, as you pointed out, further analytical studies are required to understand precisely why this approach is effective and what specific types of information loss it mitigates.
>
> ----------
>
> Regarding Question 1: We observed that performance improvements were more pronounced when applying LGPT to larger graph samples. However, since we lack a clear quantitative method to validate this, it was not included in the main text. We plan to conduct additional experiments on large-scale graph cases as soon as sufficient computational resources are available.
>
> ----------
>
> Regarding Question 2: ExplaGraph showed similar trends to SceneGraph, and for WebQSP, performance was better with 32 tokens than with 8 tokens. We attribute this to the larger graph size in WebQSP compared to the other two datasets, but as this is only an assumption, we did not mention it in the main text due to the difficulty of quantitative validation.
>
> | Prompt Token | Expla Graphs | SceneGraphs | WebQSP | Average |
> |--------------|--------------|-------------|--------|---------|
> | 1            | 88.17        | 83.51       | 70.33  | 80.67   |
> | 8            | **88.62**    | **85.19**       | 70.70  | 81.50   |
> | 32           | 80.32        | 84.21       | **70.86**  | 78.46   |
>
>
> ----------
>
> Your comments and questions align closely with the insights we gained during our research process, and we plan to investigate these areas further. However, we hope that this paper, as an interim work, will stimulate further discussions and contribute to the advancement of this research topic. We kindly ask you to reconsider your evaluation of our study.

---

> > ### Comment · Reviewer_ENaE · 2024-11-27
> > **Thank you for your response. I'd like to keep the same score.**
> >
> > Thank you for your response. I'd like to keep the same score.

---

> > > ### Author Response · Authors · 2024-11-27
> > >
> > > Feel free to let us know if you have any further questions; we will happily answer them.
> > > Thank you.

---

### Official Review · Reviewer_75jm · 2024-11-04

**Soundness:** 2
**Presentation:** 2
**Contribution:** 1
**Rating:** 3
**Confidence:** 4

**Summary:**

This paper leverages graph neural networks and large language models for task of knowledge graph question answering. Based on recent proposed techniques include graph soft prompt and query-aware graph prompting. The author proposed query-aware graph pooling to overcome the limitations of node-level and graph-level representations. In experiments, it shows competitive performance on recent proposed graph QA benchmarks in different domains.

**Strengths:**

1. The paper identifies a critical disadvantage of graph pooling method; the granularity control is either graph-level or node-level.
2. On this pain point, the proposed multiple tunrable prompt (LGPT) effecvtively imrpove the performance on benchmarks.

**Weaknesses:**

1. The novelty of the paper is questionable. As the author mentioned, recent work such as G-Retriever;Graph Token and GNP (Graph Neural Prompting) has covered most of the techniques used in the paper except the graph prompt paramters. However, the learnable graph prompt is proposed in multiple related work including [1] and supernodes (connect every node to a virtual node for pooling) in graph pooling [2] literature.

2. The proposed work re-uses most of the component of G-Retriever, which also causes my concern on cherry-picking hyperparameters given the performance improvements over G-retriever is subtle.




[1] Universal prompt tuning for graph neural networks, Neurips 2023
[2] Understanding Pooling in Graph Neural Networks,

**Questions:**

1. What's the perfomance of LGPT in figure 1 without GNN and fine-tune lanaguage model (i.e. GraphToken with LLM fine-tuning)? It would be interesting to see whether design of graph pooling is still neccessary when LLM is tunable given that GNN introduces additional parameters.

---

> ### Author Response · Authors · 2024-11-13
>
> Regarding Weakness 1:
>
> Thank you for your valuable comments. As you rightly pointed out, the use of Graph Embedding as prompts for LLMs is indeed a well-established research topic. However, our study introduces the concepts of Early Query Fusion and Learnable Pooling specifically for LLM prompts, which we believe constitute key contributions of our research.
>
> Although each module was inspired by prior works, as you and the cited research have noted, our approach to integrating these modules in the context of combining LLMs and GNNs is novel, and we consider it a primary contribution. It seems that we did not sufficiently highlight these aspects in our writing, which we will address in the revised version.
>
> ----------
> Regarding Weakness 2:
>
> To ensure a fair comparison, we minimally modified the G-Retriever while adding our proposed model. We also preserved the hyperparameters from the official code of G-Retriever. Our results show an improvement beyond the standard deviation range of G-Retriever’s average performance (1.96–5.33 standard deviations, depending on random seeds). While our sample size limits statistical testing such as t-tests, given the improvement relative to standard deviations, we believe these findings are not the result of cherry-picking.
>
> ----------
> Regarding the Question:
>
> We are unsure if we have understood your question accurately. If our answer does not align with your intent, please feel free to ask us again.
>
> In Figure 3, we report the performance of fine-tuning the LLM without a GNN. The addition of a GNN without fine-tuning the LLM resulted in higher performance compared to fine-tuning the LLM without a GNN. Furthermore, the combination of GNN addition and LLM fine-tuning showed superior performance over all other baselines.
>
> ----------
> Your thoughtful comments have greatly contributed to enhancing our research, and we are deeply grateful for this. However, we would appreciate further feedback on the rationale behind your lower rating to better address these areas. We kindly ask you to reconsider your evaluation after reviewing our response.

---

### Official Review · Reviewer_uwJ1 · 2024-11-04

**Soundness:** 2
**Presentation:** 2
**Contribution:** 2
**Rating:** 3
**Confidence:** 2

**Summary:**

The paper presents a novel approach for integrating graph representations with large language models (LLMs), addressing the critical challenge of efficient graph-text interaction. The primary contributions are twofold: (1) an early fusion mechanism that performs message passing between sub-graph node representations and query text embeddings, and (2) a learnable pooling strategy utilizing dedicated tokens (LGPT) that act as information aggregators within the graph structure.
The early fusion mechanism is particularly noteworthy as it enables direct interaction between textual and structural information at the embedding level, potentially capturing more nuanced relationships compared to traditional late fusion approaches. The authors implement this through message passing operations that allow bidirectional information flow between the sub-graph nodes and query text representations.
The learnable pooling strategy introduces fully-connected LGPT tokens that serve as dynamic information hubs within the graph. These tokens effectively aggregate information from all nodes through message passing, potentially creating a more comprehensive and adaptable graph representation. This approach appears to offer more flexibility than static pooling methods.

**Strengths:**

1. The paper introduces an innovative early fusion mechanism that addresses a fundamental challenge in graph-language modeling: the seamless integration of structural and textual information; The learnable pooling tokens (LGPT) provide a flexible and adaptive approach to graph representation, offering advantages over traditional static pooling methods.

2.The authors conduct extensive experiments across three diverse graph QA datasets, demonstrating the robustness and generalizability of their approach. The method achieves competitive performance compared to state-of-the-art baselines, while potentially offering improved computational efficiency.

**Weaknesses:**

1. The paper's scalability argument lacks sufficient comparative analysis against existing methods like G-retriever and GraphToken; The authors do not provide a detailed complexity analysis or empirical benchmarks to substantiate their efficiency claims; While the authors assert improved efficiency compared to Tian et al. 2024 (Line 210), this claim requires further scrutiny since: a). The dominant computational cost typically lies in the LLM inference; b). The relative improvement in message passing efficiency may be marginal in the overall computational pipeline; c) No concrete timing or memory usage comparisons are provided.
2. The evaluation is primarily confined to GraphQA tasks, leaving several important questions about generalization unexplored: a). The method's effectiveness on standard graph learning tasks (node classification, link prediction) remains unvalidated; b) The paper lacks a theoretical or empirical bridge between GraphQA performance and the claimed improvements in node-level and graph-level information integration. A broader evaluation across diverse graph-based tasks would strengthen the paper's contributions.
3. The hyperparameter analysis in Section 4.4 shows significant gaps in the experimental design: The LGPT token count investigation only examines extreme values (8 and 32), omitting crucial intermediate points; The impact of other critical hyperparameters (e.g., message passing steps, fusion layer configurations) is not thoroughly explored.
4. The paper should improve the methodological clarity from a). a more rigorous theoretical justification for the chosen LGPT architecture; b). Clear computational complexity analysis compared to baseline methods.

**Questions:**

1. How sensitive is the model's performance to the choice of text encoder in Equation 7?
2. Have the authors experimented with different text encoders (e.g., BERT variants, RoBERTa, T5) and observed any significant variations in performance?
3. Regarding Equation 5, how does the choice of graph encoder architecture impact the model's performance?
4. Can the authors provide case studies or visualization analysis demonstrating how LGPT addresses information loss compared to baseline methods?
5. In Equation 9, please clarify the definition and dimensionality of $S_g$
6. For Equation 10, please provide a detailed explanation of $S_p$ and its role in the architectur

---

> ### Author Response · Authors · 2024-11-13
>
> Regarding Weaknesses 1 & 4: Since GNP is a model for solving Multi-Choice QA tasks, a direct comparison with our work is not appropriate. However, we utilized GNP’s core module, the Cross-Modal Pooling Layer, to conduct an indirect performance comparison.
>
> Additionally, there seems to be a misunderstanding regarding the term "efficiency." The efficiency we described refers to representing only the information relevant to the query in a distributed manner (not concerning time or space complexity).
>
> We address the comparison of efficiency with other baselines in terms of time complexity with the same response given to reviewer M81S's question.
>
> Let n denote the number of nodes, t the number of prompt text tokens, g the number of GNN layers, and k the number of LGPTs. The time complexity of our Graph Encoder, which utilizes 3 GNNs, is therefore O(3g(n+k)). Meanwhile, the time complexity of G-Retriever and GraphToken is O(gn). Considering that k << n, the time complexity of our method and that of other baseline graph encoders remain the same at O(gn).
>
> The time complexity required for LLM computation is proportional to the square of the prompt length due to the self-attention mechanism. In our model, t+k tokens are passed in the prompt, while t+1 tokens are passed in GraphToken and G-Retriever. We set k=8 so that k << t, ensuring that the time complexity of LLM computation in our model is identical to that in other baseline models at O(t^2).
>
> ----------
>
> Regarding Weakness 2: There are two main approaches to combining LLMs and GNNs. One approach uses LLMs for solving Graph Centric Tasks, such as Node Classification or Link Prediction, while the other uses Graph Encoders for solving general NLP tasks, such as QA (https://arxiv.org/pdf/2312.02783). Our study focuses on the latter.
>
> As you suggested, investigating whether our methodology could work in the context of Graph Centric Tasks would be a valuable research direction and a worthwhile topic for future work. One of our key contributions is the development of a Graph Encoder model that can adapt to changing queries. Since Graph Centric Tasks typically involve less diverse queries than NLP tasks, further exploration is necessary to assess its applicability in that context.
>
>
> ----------
>
> Regarding Weakness 3: We regret that we could not explore a wider range of settings. Due to constraints on computational resources and time, we prioritized verifying our core modules. As you pointed out, further experiments on various hyperparameters will be essential in future studies.
>
>
> ----------
> Regarding Questions 1 & 2: We did not conduct an ablation study on the Text Encoder. Although, as you noted, experimentation with different Text Encoders could yield valuable insights, we did not pursue this as it does not critically impact the function of the core modules we propose.
>
>
> ----------
> Regarding Question 3: We focus on information transmission at the embedding level. Thus, visualizing the amount of information contained in an embedding is very challenging. At our current knowledge level, we do not have a method to visualize and compare the degree of information loss. Could you suggest a visualization approach?
>
>
> ----------
> Regarding Question 4: S_g is represented as a graph rather than a single matrix, so expressing it in dimensions may be challenging. We assume your question pertains to the dimensionality of the Node Embeddings in S_g. S_g consists of n nodes, each embedded as a vector of dimension d, resulting in an nxd Node Embedding matrix.
>
>
> ----------
> Regarding Question 5: S_p is structured similarly to S as a graph containing Node Embeddings after Pooling. However, we only project the LGPT Tokens to the LLM, so S_p information is not utilized.
>
>
> ----------
> Your insightful feedback has greatly contributed to the improvement of our research. Thank you for dedicating time to review our work. We hope that our study will be accepted, leading to further discussion. We kindly request you to reconsider your evaluation score.

---

> > ### Comment · Reviewer_uwJ1 · 2024-11-14
> > **Response**
> >
> > Thanks for the feedbacks! But they still did not resolve my questions. I will keep the same rating scores.

---

> > > ### Author Response · Authors · 2024-11-14
> > >
> > > Could you please explain in more detail which parts of our study still leave questions unresolved, separate from the scoring itself? We believe there may have been areas where we failed to communicate our ideas clearly in our writing. We would like to revise and enhance these sections to increase the completeness of our paper.
> > >
> > > Feedback on any parts that may have fallen short in clearly conveying our contributions would be incredibly helpful for us in refining this work.

---

### Author Response · Authors · 2024-11-26

## Dear Reviewers

Thank you very much for your valuable reviews, which have greatly contributed to improving the quality of our research. We sincerely appreciate your thoughtful feedback.
After thoroughly reviewing your comments, we have made the following major revisions:

1. **Section 3.3**
   We have explicitly addressed the issue of **Time Complexity**, which received significant attention, by incorporating the necessary details into the main text.

2. **Table 1**
   We have added more detailed and comprehensive descriptions of the data.

3. **Section 3.2**
   We apologize for the incorrect use of the word "efficiency" in the original text (as non-native speakers, we may have lacked precision in our word choice, and we kindly ask for your understanding).
   What we intended to convey was that embedding a small amount of information about $I_{query}$ is more **effective** than embedding all information about $I_{total}$.
   Accordingly, we have revised the wording of this section.

Thank you for your attention.

P.S.  Feel free to let us know if you have any further questions; we will happily answer them

---

### Meta-Review · Area_Chair_TBCk · 2024-12-19

**Metareview:**

Scientific Claims and Findings: This paper presents a novel approach for integrating graph representations with large language models (LLMs) for graph question-answering tasks. The key contributions include A Learnable Graph Pooling Token (LGPT) mechanism that aims to bridge the gap between node-level and graph-level projections by using learnable parameters as tokens that connect to all nodes in the graph and an Early Query Fusion technique that incorporates query context before constructing graph representations, rather than after encoding the graph.

The paper identifies and addresses a real limitation in current graph-LLM integration approaches - the tradeoff between fine-grained node-level information and efficient graph-level representations. The proposed approach shows consistent performance improvements across different experimental settings, including both frozen and fine-tuned LLM scenarios.

There is limited novelty: the core ideas build heavily on existing work: The learnable graph prompt concept appears in prior work on universal prompt tuning for GNNs. The early fusion approach using virtual nodes has precedent in the graph pooling literature. Many components are borrowed directly from G-Retriever.

The paper lacks a thorough complexity analysis comparing computational costs with baseline methods. There's limited investigation of why/how LGPT helps beyond empirical results. The hyperparameter analysis is sparse, particularly regarding LGPT token counts.

While the paper presents a well-executed study with clear empirical improvements, the primary concerns around novelty and technical depth suggest it falls slightly below the acceptance threshold. The core ideas are largely incremental combinations of existing techniques rather than fundamental advances. The missing analyses and comparisons also leave important questions unanswered about the method's efficiency and advantages. While the experimental results are positive, stronger theoretical justification and more comprehensive analysis would be needed to make this work more compelling for acceptance. The paper would benefit from addressing these gaps and potentially finding ways to differentiate its contributions more clearly from existing approaches.

**Additional Comments On Reviewer Discussion:**

During the rebuttal period, authors addressed various concerns raised by reviewers. They responded to computational complexity concerns by adding a new section analyzing and comparing time complexity with baseline methods, though empirical runtime comparisons were not included. The authors also enhanced their dataset descriptions with more comprehensive characteristics in Table 1, providing better context for evaluation, although some reviewers desired even more detailed statistics.

The terminology concerns were addressed by clarifying their use of "efficiency," explaining they meant "effectiveness" in processing query-relevant information versus total graph information. This clarification helped prevent misunderstandings about the method's advantages, though it didn't completely resolve questions about the work's novelty.

Several concerns remained unaddressed or partially addressed after the rebuttal. The similarity to existing approaches and novelty concerns weren't fully resolved. Some baseline comparisons, such as GNP, were still missing, and requests for deeper analysis of LGPT behavior and benefits weren't thoroughly addressed.

While the authors demonstrated willingness to improve the paper's clarity and technical content, particularly through the addition of complexity analysis, fundamental concerns about novelty and technical depth persisted. The clarifications and improvements, though helpful, did not significantly alter the paper's contribution level. Ultimately, despite presenting useful improvements and being well-executed, the paper did not meet the acceptance threshold due to limited novelty and incomplete technical analysis.

---

### Decision · Program_Chairs · 2025-01-22

Reject